# In Situ Gene Expression in Native Cryofixed Bone Tissue

**DOI:** 10.3390/biomedicines10020484

**Published:** 2022-02-18

**Authors:** Krisztina Nikovics, Cédric Castellarin, Xavier Holy, Marjorie Durand, Halima Morin, Abdelhafid Bendahmane, Anne-Laure Favier

**Affiliations:** 1Imagery Unit, Department of Platforms and Technology Research, French Armed Forces Biomedical Research Institute, 91223 Brétigny-sur-Orge, France; cedric.castellarin@intradef.gouv.fr (C.C.); anne-laure.favier@intradef.gouv.fr (A.-L.F.); 2Department of Platforms and Technology Research, French Armed Forces Biomedical Research Institute, 91223 Brétigny-sur-Orge, France; xavier.holy@intradef.gouv.fr; 3Osteo-Articulary Biotherapy Unit, Department of Medical and Surgical Assistance to the Armed Forces, French Armed Forces Biomedical Research Institute, 91223 Brétigny-sur-Orge, France; marjorie1.durand@intradef.gouv.fr; 4National Research Institute for Agriculture, Food and the Environment (INRAE), Institute of Plant Sciences Paris-Saclay (IPS2), University Paris-Saclay, 91400 Orsay, France; halima.morin@inrae.fr (H.M.); abdelhafid.bendahmane@inrae.fr (A.B.)

**Keywords:** cryofixation, bone, in situ hybridization, hybridization chain reaction (HCR), macrophage

## Abstract

Bone is a very complex tissue that is constantly changing throughout the lifespan. The precise mechanism of bone regeneration remains poorly understood. Large bone defects can be caused by gunshot injury, trauma, accidents, congenital anomalies and tissue resection due to cancer. Therefore, understanding bone homeostasis and regeneration has considerable clinical and scientific importance in the development of bone therapy. Macrophages are well known innate immune cells secreting different combinations of cytokines and their role in bone regeneration during bone healing is essential. Here, we present a method to identify mRNA transcripts in cryosections of non-decalcified rat bone using in situ hybridization and hybridization chain reaction to explore gene expression in situ for better understanding the gene expression of the bone tissues.

## 1. Introduction

Vertebrate bone is a dynamically changing tissue that constantly adapts throughout life. For successful bone healing, coordinated cross talk is needed between inflammatory and bone-forming cells [1,2,3,4,5]. Nowadays, the exact mechanisms of bone regeneration remains to be elucidated [6,7,8,9]. Bones and bone marrow contain different types of macrophages: (i) erythroid island macrophages; (ii) hematopoietic stem cell macrophages; and (iii) osteoclasts [10]. Macrophages play an important role both in osteoblast-mediated bone formation [9] and in osteoclast development [10]. In addition, the newly-discovered osteal macrophages, so called “osteomacs”, have a fundamental role during bone regeneration [6,9,11,12]. The exact role of these cells is still under study. The cytokines and other soluble factors secreted by macrophages can induce the bone formation in vitro [2,5,9,13]. Cytokines are the critical actors in coordinating an efficient repair of damaged bone tissue [14]. Examination of cytokine expressions during bone regeneration is essential for establishing new diagnostic and therapeutic approaches for bone tissue repair. Protein expression analysis of bone cells mostly uses immunofluorescence technique methods to detect proteins in situ [15]. In spite of immunolabeling being very convenient and well reproducible, there are some disadvantages, such as non-specific labeling with certain antibodies [16,17]. Additional difficulty is that cytokines are usually secreted, so the identification of cells producing this peptide or protein is problematic [16].

In contrast, in situ hybridization (ISH) is one of the most suitable methods to investigate localization of gene expression in situ and based on the detection of mRNA product of genes involved in protein translation [16,18,19,20,21]. The basis of this technique is that complementary RNA and DNA sequences form hybrids with one another by hydrogen bonding. ISH is a very powerful technique; however, the probe design is complex, and the different steps are fastidious, needing a high level of optimization. An additional problem is that the conservation of the RNA as mRNA is very sensitive to degradation and RNase enzymes can be found everywhere [20].

The digoxigenin technique (in situ-DIG) is the most commonly used non-radioactive IHS method [22,23]. It is highly sensitive, but unfortunately only allows the analysis of a single gene in a single sample. In the 1980s, the fluorescence in situ hybridization (FISH) technique was published [24]. This method has the advantage of allowing the simultaneous analysis of several genes in the same sample, but is only suitable for the analysis of highly expressed genes [25]. Recently, a new method has been published in which in situ hybridization is coupled with the detection by hybridization chain reaction (in situ-HCR) [21,26,27,28]. One of the main advantages of this technique is its high sensitivity that makes it suitable for testing low-expression genes. In addition, several genes can be analyzed simultaneously. However, in situ-HCR is less efficient than the in situ-DIG method [25].

We found that fixation in buffered formalin, decalcification in EDTA and embedding in paraffin is a good compromise, as it provides not only a good morphology and excellent conditions for immunohistochemistry but also allows DNA- and RNA-based molecular studies [15,29,30] (Table 1).

Cryosectioning of hard tissue has been introduced several decades ago [31] and was optimized by the tape technique described by Kawamoto et al. [32,33,34]. The main advantage of this system is that there is no fixation and embedding before section preparation, so it is much faster than the conventional method and more useful for in situ hybridization (Table 1). As mRNA is very sensitive to degradation, the challenge resided in the ability to cut cryofixed bone tissue and to preserve mRNA for analysis. To overcome these limitations, cryosectioning of the bone of rat was combined with ISH on the entire femur together with the muscle.

## 2. Materials and Methods

### 2.1. Rat Animal Model

All experiments were approved by the IRBA Institutional Animal Care and Use Committee (protocol 65 DEF_IGSSA_SP). Interventions were carried out in an accredited animal facility. 8-week-old (200 g average weight) male Sprague Dawley rats (Charles River Laboratories, Freiburg, Germany) were housed individually in cages, in a temperature- and light-controlled environment, with food and water ad libitum. Before collecting the femurs with muscles, animals were euthanized at 12 weeks old with an overdose of sodium pentobarbital (150 mg/kg) administrated intraperitoneally.

### 2.2. Slides Coating

Slides were manually coated in two steps using coating and pretreatment solutions (Leica Microsystems, Richmond, IL, USA). Slides were first pretreated with A solution (5 mL of A buffer concentrate (39475270, Leica, Wetzlar, Germany), 1.25 mL of 0.1 M acetic acid and 25 mL of acetone in 500 mL distilled water) (Figure 1A). After 30 min in dark, the A solution was completed with acetone (q.s.p. 500 mL), filtered and stored until 6 months at 4 °C in a dark bottle. Slides were immersed 3 times in A solution and pulled out diagonally to avoid streaks. Slides were kept overnight at RT or warmed at 90 °C for 5 min before the coating process. A total of 15 μL of B solution (39475271, Leica) was coated on pretreated slides to obtain a thin and homogenous coating (Figure 1B,C). The surface of the coating was adapted to match with the cryosection surface (Figure 1D).

### 2.3. Embedding and Cryosectioning of Entire Femur of the Rat

RNase-free instruments, materials and buffers were used to collect bone samples. After euthanasia of the rat, the whole femur was cleaned rapidly and a part of the muscles around the bone was kept. The femur was placed at the bottom of the embedding mold and covered with cryomount medium (CM) (00890-EX, HistoLab, Askim, Norvege). Samples were snap-frozen with 2-methylbutan cooled in liquid nitrogen to obtain a block (Figure 1E). When entirely frozen, the sample was transferred on dry ice to −80 °C, wrapped in foil aluminum. Then, it was stored at −80 °C until further processing. A cryostat (Cryostat FSE Shandon, Thermo Electron Corporation, San Diego, CA, USA) with a Leica CryoJane (9194701, Leica) system was used for cryosectioning. The block was fixed with cryomount medium (Appendix A). Tools were precooled within the cryostat to avoid warming up the sample during block trimming (Figure 1F). The surface of the adhesive film (39475214, Leica) was adapted to the surface of the block (Figure 1G) and CryoJane tape transfer system was applied (39475205, Leica) to obtain high-quality sections.

The following points are crucial to the success of the experiment:

Briefly: (i) Positioning the adhesive film on the surface of the block (Appendix A). (ii) Applying the roller while exerting a certain force to improve the adhesion of the film (Appendix A). (iii) Cutting the sample slowly and uninterruptedly (Appendix A). Obtaining 5 µm tissue sections on the adhesive film (Figure 1H and Appendix A). (iv) Transferring the section from the film to a standard histological slide, manually pretreated with Leica’s A and B solutions (Appendix A). (v) Fixation of the section on the coated side of the slide by CryoJane UV flash system (Appendix A). (vi) Removal of the adhesive film (Appendix A) and optimization the transfer of the cryo-section (Appendix A). Four points are decisive to ensure good quality of the cryosections: (1) The CM block is very important for a good section (the tissue alone does not adhere well to the film). The film must be in contact with the CM and with the tissue. (2) The entire surface of the adhesive film must be in contact with the CM block. The film must not be wider than Section 3 and Section 4. Very important is to hold the bottom of the film with a pair of pliers when cutting—this avoids heating the film which must remain cold and is safer (Figure 1M).

### 2.4. Histological Staining

Hematoxylin and phloxin (HP) staining was performed as followed: the sections were incubated in several successive baths: 40 s in hemalum (11487, Merck, Darmstadt, Germany) buffer (0.2 g hemalum, 5 g aluminum potassium sulfate in 100 mL distilled water), 3 min in water, 30 s in phloxin (15926, Merck) buffer (0.5 g phloxin in 100 mL distilled water), 1 min in water, 2 min in 70% ethanol, 30 s in 95% ethanol, 1 min in 100% ethanol, 1 min in 100% ethanol. At the end, nuclei were colored in blue and cytoplasm in pink (Figure 1I–K).

Von Kossa staining was performed as followed: Sections were rinsed with distilled water and incubated for 30 min in the dark with silver nitrate solution (1 g silver nitrate in 100 mL distilled water) at room temperature. After washes with distilled water, the sections were incubated under UV for one hour (sections should be covered with distilled water). Finally, the short passage in 95% and 100% ethanol, followed by xylene, were carried out. For the good conservation of the cryo-section, Eukitt mounting solution was used (Figure 1L).

### 2.5. Immunofluorescence

Sections of rat femur were fixed in paraformaldehyde (PFA (P6148, Sigma, Lezennes, France); 4% (*w*/*v*) in PBS (Phosphate-Buffered Salin without Ca and Mg, GAUPBS0001, Eurobio, Les Ulis, France)). After three washes in PBS, the sections were permeabilized for 15 min with PBS containing 0.5% Triton X100 (*v*/*v*). The non-specific binding sites were blocked with Emerald Antibody Diluent (Sigma 936B-08) for 1 h. The sections were incubated overnight at 4 °C with the primary rabbit anti-CD68 (ab125212, Abcam, Amsterdam, The Netherlands) antibody at 1:1000 dilutions. Then they were washed in PBS and incubated with the secondary anti-rabbit Alexa Fluor 488 (A-21206, Thermo Scientific, Villebon sur Yvette, France) antibody at 1:500 dilution for 2 h at room temperature. Finally, sections were washed in PBS for 20 min and mounted using a Fluoroshield mounting medium with DAPI (Abcam, ab104139). Fluorescence was detected using an epifluorescence microscope DM6000 (Leica, Germany) equipped with monochrome and color digital cameras.

### 2.6. In Situ-DIG Hybridization

For in situ-DIG hybridization, a couple of primers were chosen to get an approximately 1000 bp (β-actin (944 bp, CD68 1059 bp) PCR amplicon (Appendix A). β-actin and CD68 DNA primers were designed using ApE software (Appendix A). Two other primers, including 62 bp of the PCR product, were designed to recognize the T3 and T7 promoter sequences to perform the in vitro transcription (Figure 2). DNA oligos were synthesized by Eurogentec and were dissolved in ddH2O and stored at −20 °C.

#### 2.6.1. cRNA Probes for In Situ-DIG Hybridization

Frozen femur samples were homogenized in liquid nitrogen. RNA was isolated using RNeasy Fibrous Tissue mini kit (HB-0485, Qiagen, Courtaboeuf, France) according to the manufacturer’s recommendations. RNA extracts were eluted with 20 µL RNase-free water. Reverse transcription (RT) was performed with oligo(dT) primers following the instructions of the Sensiscript transcription kit (205211, Qiagen). Reactions were carried out using 50 ng of RNA with 10 µM oligo(dT) primers, RNase inhibitor (2 IU) and Sensiscript reverse transcriptase. cDNA was synthesized at 37 °C during 60 min. This cDNA library was used for the amplification of the β-actin templates for in vitro transcription. These templates contained the T3 and T7 promoters. The cRNA labeling was generated by in vitro transcription with T3 and T7 RNA polymerases, both in antisense and sense direction. Sense probes were used as controls. The PCR fragments were purified by agarose gel electrophoresis and specific bands were isolated using PCR Clean-Up kit (740,609.10, Macherey-Nagel, Hoerdt, France).

Next, RNA was labeled using an in vitro transcription Kit (P1450, Promega, Charbonnières les Bains, France) according to manufacturer’s recommendations. RNA was labeled with digoxigenin, by addition of a modified nucleotide, DIG-11-UTP. Transcription was carried out in buffer containing dNTPs, DIG-11-UTP, RNase inhibitor and RNA polymerase at 37 °C for 1 h. DNAse was then added and the mix further incubated for 30 min at 37 °C to degrade DNA.

A total of 10 µL (10 mg mL^−1^) transfer RNAs (1010945001, Roche, Boulogne-Billancourt, France) were added to the mix and the probes were precipitated with 10 M ammonium acetate (A1542, Sigma) and 100% cold ethanol overnight at −20 °C. After centrifugation at 15,000× *g* at 4 °C for 30 min, the pellet was rinsed with 70% cold ethanol. To improve probe penetration for probes longer than 500 nucleotides, hydrolysis of the probe is needed. For hydrolysis, the probe was suspended in 50 µL carbonate buffer (120 mM Na_2_CO_3_, 80 mM NaHCO_3_, pH 10.2) and incubated at 60 °C for 55 min. The reaction was stopped with a buffer containing 10 µL of 10% acetic acid, 12 µL 3 M sodium acetate (pH 4.8) and 312 µL 100% cold ethanol and incubated at −20 °C for at least 30 min. After centrifugation, the pellet was washed with 70% cold ethanol and suspended in 50% RNase-free formamide.

The effectiveness of labeling was analyzed by dot-blots. cRNA were dot-blotted on a nitrocellulose membrane (88018, Thermo Fisher Sci., Illkirch, France), and detected with an anti-digoxigenin (11093274910, Roche) DIG-specific antibody. From each serial dilution of the probes, 1 μL was spotted on a nitrocellulose membrane, dried and UV-crosslinked for 1 min. To prevent nonspecific antibody binding, the membrane was blocked with 1% bovine serum albumin (BSA; GAUBSA01, Eurobio, Les Ulis, France) in 100 mM Tris pH 7.5 and 150 mM NaCl (Tris-NaCl) buffer for 15 min. Afterwards the membrane was incubated for 30 min with anti-DIG antibody at 1:2000 dilutions in BSA/Tris-NaCl and then washed three times for 5 min in BSA/Tris-NaCl and once with Tris-NaCl. Finally, the membrane was stained with an NBT kit (NBT/BCIP; S3771, Roche) according to the manufacturer’s recommendations.

#### 2.6.2. Fixation and Pretreatment of Sections for In Situ-DIG Hybridization

All the following steps were performed under a laminar flow cabinet and under RNase-free conditions. The tissues were fixed in 4% (*w*/*v*) in PBS/paraformaldehyde (PBS, *w*/*o* Ca and Mg, GAUPBS0001, Eurobio; PFA, P6148, Sigma) for 30 min, and then treated with 100% methanol for 15 min and air-dried. Sections were incubated in 0.125 mg mL^−1^ Proteinase K (P2308, Sigma) in 200 mL 100 mM Tris pH 7.5 and 50 mM EDTA buffer for 10 min at 37 °C to degrade the proteins and to improve the probes’ access to the target mRNA. Proteinase K reaction was stopped with 0.2% glycine (G7126, Sigma) in 1X PBS. Then sections were treated with 0.5% acetic anhydride (A6404, Sigma) in triethanolamine solution (0.1 M pH 8) to avoid non-specific hybridization. The sections were then washed twice for 2 min with PBS and dehydrated with successive baths of saline solution and ethanol: 30 s in 30% ethanol, 0.85% NaCl buffer; 30 s in 50% ethanol, 0.85% NaCl buffer; 30 s in 75% ethanol, 0.85% NaCl buffer; 30 s in 85% ethanol, 0.42% NaCl buffer; 30 s in 96% ethanol; 30 s in 96% ethanol; 1 min in 100% ethanol. Slides were then stored at −20 °C.

#### 2.6.3. Prehybridization and Hybridization for In Situ-DIG

The sections were pre-hybridized for 2 h at 45 °C in a pre-hybridization buffer (50% formamide (GHYFOR0402, Eurobio), 0.5× sodium chloride citrate (SSC) (GHYSSC007, Eurobio) buffer, 50 µg mL^−1^ heparin (H3393, Sigma), 100 µg mL^−1^ transfer RNA and 0.1% (*v*/*v*) Tween 20 (822184, Merck). Finally, the sections were incubated overnight at 45 °C with the RNA probes (2 µL probe in 200 µL hybridization buffer (50% formamide, 100 µg mL^−1^ transfer RNA, 7.5% (*v*/*v*) Tween 20, 8.5% NaCl, 20% dextran sulfate (Eurobio GHYDEX000T) and 2.5× Denhardt’s Solution (50× stock, D2532, Sigma)), which were previously denaturized for 2 min at 80 °C in the hybridization buffer.

Non-specific hybrids were dissociated with following washes: 30 min in 0.1× SSC + 0.5% SDS at 45 °C, 2 h in 2× SSC + 50% formamide at 45 °C, 5 min in NTE (0.5 M NaCl, 10 mM Tris pH 8, 1 mM EDTA) at 45 °C, 30 min in NTE + 10 mg ml-1 Rnase A (10109169001, Roche) at 37 °C, 1 h in 2× SSC + 50% formamide at 45 °C, 2 min in 0.1× SSC at 45 °C and finally 15 min in PBS at RT.

#### 2.6.4. Detection for In Situ-DIG

Immunodetection of the DIG-labeled probes was performed using an anti-DIG antibody coupled to alkaline phosphatase, as described by the manufacturer (11093274910, Roche). For the immunological detection step, the sections were incubated in a first buffer (0.5% Blocking reagent (1110961176001, Roche) in 100 mM Tris pH 7.5 and 150 mM NaCl) for 1 h and in a second one (1% BSA in 0.5% (*v*/*v*) Triton X100, 100 mM Tris pH 7.5 and 150 mM NaCl) for 1 h to block unspecific sites. Sections were incubated with anti-digoxygenin antibody 1:1250 for 1 h, and then washed three times for 20 min with 1% BSA in 0.5% (*v*/*v*) Triton X100, 100 mM Tris pH 7.5 and 150 mM NaCl solution and next incubated for 15 min in the same solution without BSA. Finally, this solution was replaced with the last buffer (100 mM Tris pH 9.5, 100 mM NaCl and 50 mM MgCl_2_) for 15 min.

Staining was initiated at alkali pH. The sections were incubated for 1–2 days in a buffer containing 337 µL BCIP (5-bromo-4-chloro-3-indolyl-phosphate) and 225 µL NBT (Nitroblue tetrazolium chloride) in 50 mL solution (100 mM Tris pH 9.5, 100 mM NaCl and 50 mM MgCl_2_) until a blue precipitate adhering to the sections was formed. The reaction was stopped by adding a stop solution (10 mM Tris pH 7.5 and 5 mM EDTA) for 10 min. The DIG sections were observed with an epifluorescence microscope DM6000 (Leica, Germany) equipped with monochrome and color digital cameras while the HCR sections were observed with a confocal microscope (LSM700, Zeiss, Dresden, Germany).

### 2.7. In Situ-HCR Hybridization

The HCR protocol of Choi and colleagues (2014, 2016) was performed with some modifications as described below to enhance mRNA localization in the femur of the rat [21,26].

#### 2.7.1. β-Actin Probe Design for In Situ-HCR Hybridization

The probes were designed using ApE software. For each gene, five probes were designed. The entire gene sequence was used to localize the introns, and probes were designed exactly at the boundaries between two exons. This approach increased the capacity of probes to hybridize with the mRNA and not with the genomic DNA. A specific additional sequence was included to interact with the hairpin coupled with the fluorophore [21]. DNA oligos were synthesized by Eurogentec. Details of probe sequences are described in Appendix A. All oligos were dissolved in ddH2O and stored at −20 °C.

#### 2.7.2. Fixation and Pretreatment of Sections for In Situ-HCR Hybridization

This process was common for both DIG and HCR in situ hybridization methods.

#### 2.7.3. Prehybridization and Hybridization for In Situ-HCR

The sections were pre-hybridized for 10 min at RT in a hybridization buffer (50% formamide, 5× SSC, 9 mM citric acid pH 6, 50 µg mL^−1^ heparin, 1× Denhardt’s Solution, 0.1% (*v*/*v*) Tween 20 and 10% dextran-sulfate). Previously, the hybridization probes (2 pmol per slide) were denatured for 2 min at 80 °C. Finally, the sections were incubated in a hybridization buffer together with probes overnight at 45 °C. Nonspecific hybrids were dissociated with the following washes: 30 min in 0.1× SSC + 0.5% SDS at 45 °C, followed by 2 h in 2× SSC + 50% formamide at 45 °C, and then 2 min in 0.1× SSC at 45 °C and finally 15 min in PBS at RT.

#### 2.7.4. Detection for In Situ-HCR

Sections were first incubated for 2 h at RT with an amplification buffer (5× SSC, 0.1% (*v*/*v*) Tween 20, 10% dextran-sulfate and 100 µg mL^−1^ salmon sperm ADN) and subsequently for 12 to 16 h with the DNA hairpins marked with a fluorophore (Alexa Fluor488) (diluted in amplification buffer, as described previously). The hairpins were previously heated at 95 °C for 90 s and cooled to RT for 30 min. The sections were then washed 2× 30 min in 5× SSCT (5× SSC and 1% (*v*/*v*) Tween 20) and 5 min with 5× SSC without Tween at RT.

### 2.8. Microspectrofluorimetry

Emission fluorescence spectra was measured between 460 and 650 nm (5 nm bandwidth) with a Leica TCS SP8 Confocal Microscope. Cyan fluorescence was excitated at 488 nm. Washes: 30 min in 0.1× SSC + 0.5% SDS at 45 °C, followed by 2 h in 2× SSC + 50% formamide at 45 °C, and then 2 min in 0.1× SSC at 45 °C and finally 15 min in PBS at RT.

## 3. Results and Discussion

The particular mechanism of bone regeneration is under active examination. However, preparing a histology section from bone is quite difficult due to the mineralization of this tissue. Mineralized tissues should be decalcified from 1 to 2 weeks before embedding and sectioning. Decalcification can alter the antigenicity of certain proteins and can cause degradation of RNA molecules. Therefore, immunohistochemistry and ISH are less possible on the sections [35,36,37]. The success of the ISH technique realization extremely depends on the quick preparation of good-quality bone tissue sections. The aim of our work was to develop a new approach for obtaining high-quality undecalcified bone sections applicable to various ISH analyses.

Because ISH is based on mRNA analysis, it is essential to develop a procedure for maintaining an RNase-free lab (RNase-free instruments and materials, wear gloves and work quickly to avoid storing samples at RT) so as to conserve mRNAs of sufficient quality and quantity for subsequent analyses. Rapid techniques without any prolonged aqueous phase steps are crucial to prevent RNA degradation. Manual coating of slides greatly helped to optimize sample attachment to the slide during in situ hybridization (Figure 1A–D), with the area of coating to be adjusted to the sample area (Figure 1D).

First, the entire femur together with muscle was embedded in a cryo-embedding medium, frozen and trimmed (Figure 1E). All tools were maintained cold into the cryo-bar (Figure 1F). Several shapes of adhesive film were prepared to fix equivalent surfaces of cryosections (Figure 1G). Thin (5 µm) cryosections were cut (Figure 1H and Appendix A), stained first with HP (Figure 1I–K) and then with von Kossa medium (Figure 1L) to confirm the capacity of the cryosection technique to retain morphological structures and mineralization of the bone. To optimize the cryosectioning and transfer, four technical points were described in detail (Figure 1M).

Subsequently, the bone sections were analyzed using a conventional immunofluorescence technique (with anti-CD68 antibody) to identify the macrophages (Figure 3). One of the most widely used markers for the analysis of monocytes/macrophages is the Cluster of Differentiation (CD) CD68 protein [38,39]. Although weakly expressed in other mononuclear phagocyte cells, this glycoprotein is highly expressed in macrophages. Very weak expression can be detected in other non-hematopoietic cells (mesenchymal stem cells, fibroblast, endothelial and tumor cells) [40,41]. Monocytes/macrophages were detected in the bone marrow (Figure 3A), and in the interface between periosteum and cortical bone (Figure 3B). Osteoclast and macrophages have similar origins and both produce CD68 protein [10,42,43]. We identified the presence of osteoclasts as multinucleated cell expressing CD68 protein (Figure 3B). No expression was identified in the negative control (Figure 3C,D).

Next, ISH with a digoxigenin-labeled probe (in situ-DIG) was performed. Specific probes were designed and synthetized to target the mRNAs of β-actin in bone, a ubiquitous protein with a strong expression in almost every cell [44,45,46], then a 944 bp complementary RNA (cRNA) was generated (Figure 2, Appendix A). An mRNA probe was chosen because the RNA–RNA interaction is more efficient than the RNA–DNA interaction [20].

The cellular localization of the β-actin was analyzed by the in situ-DIG method (Figure 4). When the RNA probe used for hybridization was the same sense (not complementary) as the mRNA (negative control), no labeling was observed (Figure 4A,C), whereas the expression of β-actin was very intense in the bone marrow (Figure 4B) and in the periosteum (Figure 4D).

In situ hybridization coupled with hybridization chain reaction detection (in situ-HCR) was chosen because this approach is more sensitive than fluorescence in situ hybridization (FISH) [18,47,48] and allows identification of several genes at the same time [16,21,28,49]. Indeed, β-actin expression was analyzed in a rat bone animal model, using a DNA probe linked to a fluorophore instead of an enzyme. In the absence of a probe (negative control), no labeling was observed (Figure 4E,G), whereas a strong labeling was detected both in bone marrow (Figure 4F) and in the periosteum (Figure 4H). DAPI was used as nuclear counterstain.

To go further with our investigation, the identification of macrophages we used both the in situ-DIG and HCR techniques. Macrophages were identified based on the expression of CD68 mRNA (Figure 5). CD68 mRNA expression was strong in the bone marrow (Figure 5A,C) and in the interface between the periosteum and cortical bone (Figure 5B,D) with both in situ techniques.

Autofluorescence of bone tissue was previously described [50,51,52]. To confirm that the detected fluorescence resulted from an in situ-HCR signal and did not derive from nonspecific binding of an Alexa Fluor 488 molecule or from autofluorescence, hybridization was evaluated by microspectrofluorimetry (Figure 6). The emission spectra of the fluorescence peaked at 520 nm, corresponding to Alexa Fluor 488 fluorophore spectra.

ISH analysis is an important technique used in order to understand the molecular mechanism of bone tissue regeneration. Our aim was to analyze the section of the entire rat femur. Paraffin embedding was not appropriate in our case, because the muscle slowed down the penetration of the decalcifying solution (EDTA) into the bone. Thus, the decalcifying treatment was very long (around 5 weeks), which prevented the ISH on these samples (data not shown). We developed an improved version of the CryoJane tape transfer system to prepare cryosections of undecalcified rat femurs with good bone tissue morphology and applicable for ISH analyses. We demonstrated that in situ-HCR is a promising new technique for visualizing macrophages and studying the expression of different genes in bone tissue.

## 4. Conclusions

Bone healing is very complex process. For proper regeneration, dynamic interplay between external and internal signals is essential. Cytokines of the macrophages and other cellular factors act at diverse times, and have indispensable functions during repair. In order to make progress in bone regeneration understanding, the development of new tools is essential. These tools will provide opportunities to explore in situ the spatial actors involved in inflammation and bone tissue regeneration. Indeed, the application of distinct ISH approaches can bring new comprehension to bone gene expression and tissue regeneration. Combination of cryofixation with ISH techniques is a relevant approach to study the molecular and spatial biological mechanisms of bone regeneration and provides advanced perspectives in the field of regenerative medicine to induce bone regeneration for developing new treatments.

## Figures and Tables

**Figure 1 biomedicines-10-00484-f001:**
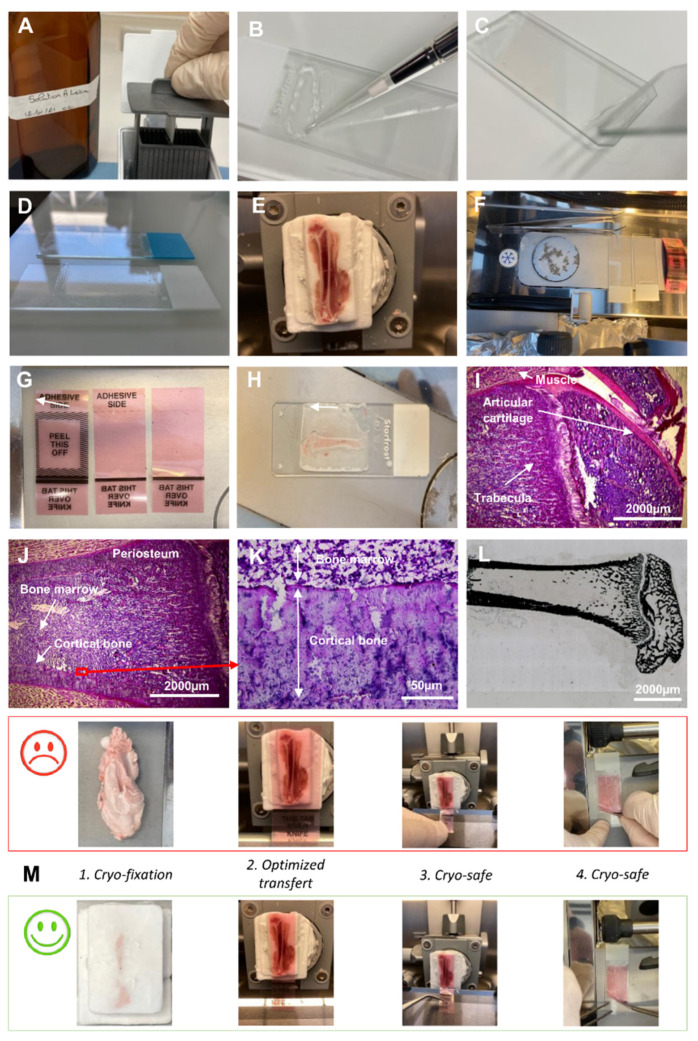
Cryo-fixation and cryo-section steps. Slides were manually coated with (**A**) A and (**B**) B buffer. (**C**) Thin B buffer layer on the slide surface. (**D**) Adjusted coating of the manually prepared slide compared with the manufactured one. (**E**) Entire cryo-embedded and trimmed femur of rat. (**F**) Cryo-bar to precool slides, adhesive film and tweezers. (**G**) Several surface size of adhesive film. (**H**) Thin (5 µm) cryo-section of femur deposed on a pre-coated slide. (**I**,**J**) Histological coloration of femur stained with Hemalin–phloxine–saffron. (**K**) Expanded view: high magnification image of the area within the red rectangle in image (**J**). (**L**) Von Kossa staining of the mouse and rat bone. (**M**) Upper panel: technical practices to avoid; down panel: to improve the result in comparison with the appropriate practices.

**Figure 2 biomedicines-10-00484-f002:**
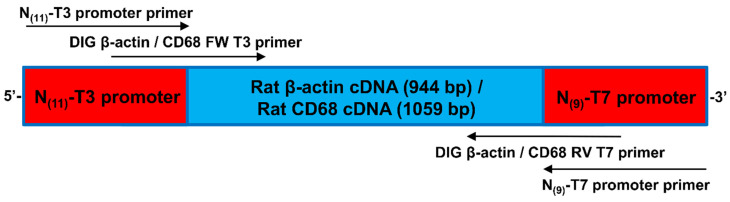
Schematic representation of the PCR product (together with primer localization and orientation) used for in vitro transcription.

**Figure 3 biomedicines-10-00484-f003:**
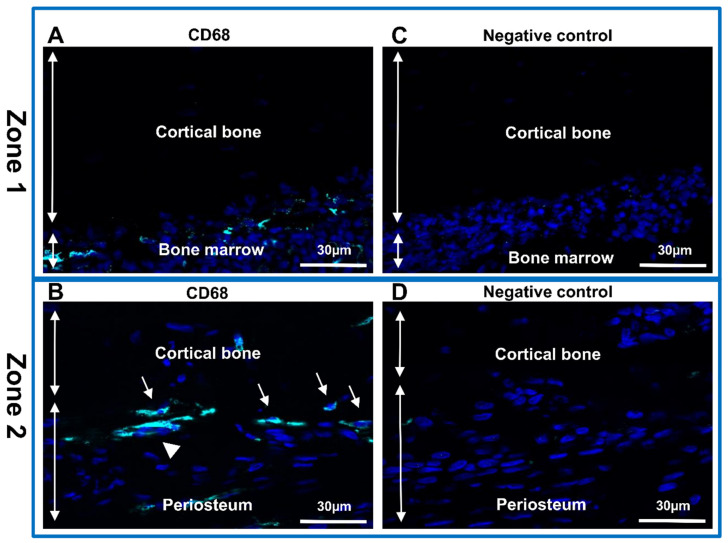
Identification of macrophages and osteoclast in rat femurs. Immunolabeling with anti-CD68 antibody in sections of two representative zones of the sample. (**A**,**C**) Zone 1-cortical bone and bone marrow; (**B**,**D**) Zone 2-periosteum and cortical bone. (**A**,**B**) Anti-CD68 (Alexa488, turquoise fluorescence), labeling the macrophages and osteoclasts. (**C**,**D**) Negative control. Nuclear staining with DAPI (blue fluorescence). Thin arrow: macrophages; arrowhead: osteoclast.

**Figure 4 biomedicines-10-00484-f004:**
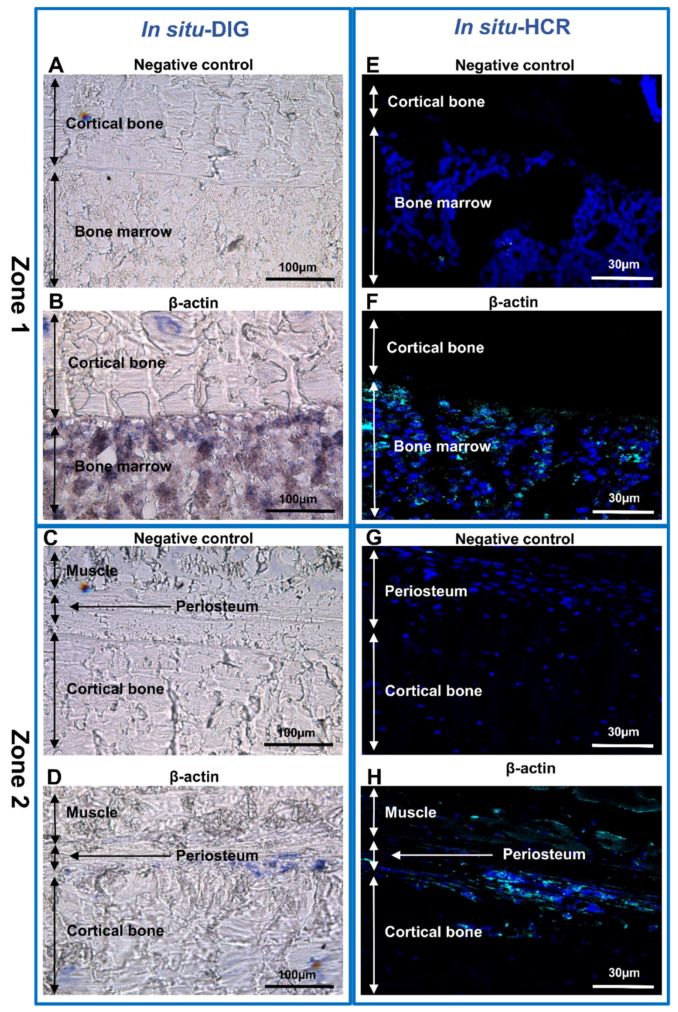
In situ hybridization analysis of non-decalcified rat bone section. (**A**–**H**) β-actin expression in sections of two representative zones of the sample. (**A**,**B**,**E**,**F**) Zone 1-cortical bone and bone marrow; (**C**,**D**,**G**,**H**) Zone 2-muscle, periosteum and cortical bone. (**A**–**D**) In situ-DIG; (**E**–**H**) in situ-HCR. (**A**,**C**,**E**,**G**) Negative control. (**B**,**D**,**F**,**H**) Expression of β-actin mRNA in non-decalcified rat bone.

**Figure 5 biomedicines-10-00484-f005:**
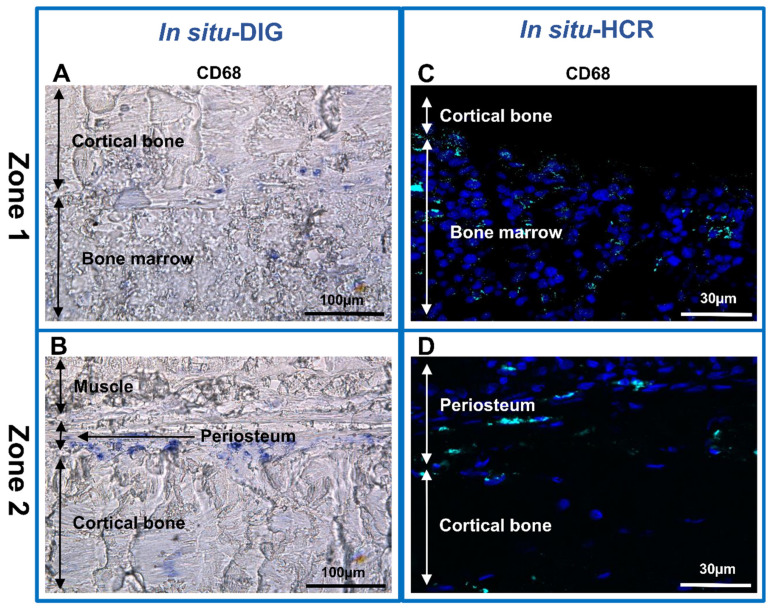
In situ hybridization analysis in non-decalcified rat bone section. (**A**–**D**) CD68 mRNA expression in section of two representative zones of the sample. (**A**,**C**) Zone 1-cortical bone and bone marrow. (**B**,**D**) Zone 2-muscle, periosteum and cortical bone. (**A**,**B**) In situ-DIG; (**C**,**D**) in situ-HCR.

**Figure 6 biomedicines-10-00484-f006:**
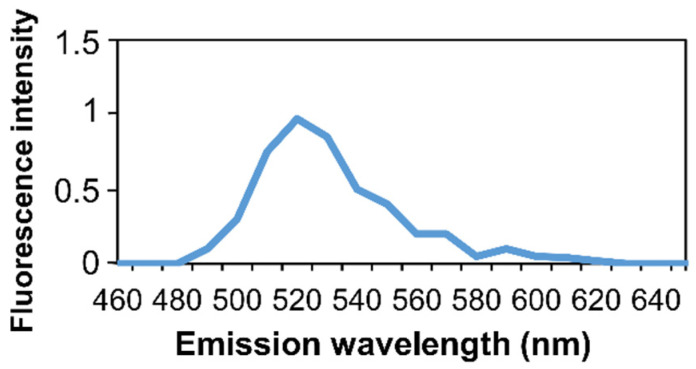
Microspectroscopy analysis of non-decalcified rat bone section. In situ microspectroscopy analysis of the Alexa Fluor 488 fluorescence.

**Table 1 biomedicines-10-00484-t001:** Comparison of the different approaches.

	Resin (R)/Paraffin (P)-Section	Cryo-Section
Section preparation	Long method	Short method
Toxic substances	Long period	Short period
Size of the section	5 μm	5 μm
Quality of morphology	Good	Medium
Application	Histological staining (R, P) Immunolabeling (P) In situ hybridization (P, medium sensitivity)	Histological staining Immunolabeling In situ hybridization (strong sensitivity) Laser microdissection

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
