# Peer review of "In Situ Gene Expression in Native Cryofixed Bone Tissue"

_biomedicines, 2022, doi:10.3390/biomedicines10020484_

Round 1

Reviewer 1 Report

The authors described application of in situ hybridization (ISH)/hybridization-chain-reaction (HCR) on cryosections of bone. Although slide coating requires commercially available solutions with unknown composition, and cryosectioning requires specific adhesive films, the ISH/HCR method is clearly written and is certainly attractive to many bone biologists. However, there are two specific points which the authors should consider.

Specific points.

1. The authors used beta-actin as an example, which is disappointing. The authors should use genes expressed in osteoblasts/osteocytes or osteoclasts. Showing double staining in osteoblasts and osteoclasts would strengthen the manuscript dramatically.

2. The nature of the adhesive film should be provided. If commercially available, please provide the information. What is the difference between the Kawamoto’s Film Method and this? At least the authors should quote the Kawamoto reference.

Kawamoto T, Kawamoto K. Preparation of Thin Frozen Sections from Nonfixed and Undecalcified Hard Tissues Using Kawamoto's Film Method (2020). Methods Mol Biol. 2021;2230:259-281. doi: 10.1007/978-1-0716-1028-2_15. PMID: 33197019.

Author Response

Dear Reviewer 1,

The Reviewer 1 comments greatly helped to improve our manuscript. We have modified the manuscript by taking into account each comment and remark.

The authors described application of in situ hybridization (ISH)/hybridization-chain-reaction (HCR) on cryosections of bone. Although slide coating requires commercially available solutions with unknown composition, and cryosectioning requires specific adhesive films, the ISH/HCR method is clearly written and is certainly attractive to many bone biologists. However, there are two specific points which the authors should consider.

Specific points.

  1. The authors used beta-actin as an example, which is disappointing. The authors should use genes expressed in osteoblasts/osteocytes or osteoclasts. Showing double staining in osteoblasts and osteoclasts would strengthen the manuscript dramatically.

The comment has been taken into account. Additional experimental data have been included (Fig. 2, Fig. 3, Fig. 5, Fig. 6, Table 1 and Table S1). The manuscript has been modified to improve the message of the paper (see lines 37-43, 135-143, 240-243, 287-295, 312-319, 329-330).

The expression of beta-actin is uniform in almost every cells. Therefore, beta-actin is a useful positive control of in situ hybridization and it is frequently used as a positive control in publications.* Its strong and “ubiquitous” expression is an advantage.

*F Y Gan, G D Luk and M S Gesell. (1994) Nonradioactive in situ Hybridization Techniques for Routinely Prepared Pathology Specimens and Cultured Cells. Journal of Histotechnology Volume 17 313-319 | Published online: 29 Nov 2013.

K L Taneja and R H Singer. (1990) Detection and localization of actin mRNA isoforms in chicken muscle cells by in situ hybridization using biotinated oligonucleotide probes. J Cell Biochem, 1990 Dec;44(4):241-52. doi: 10.1002/jcb.240440406.

T C Hoock, P M Newcomb, and I M Herman. (1991) Beta-Actin and its mRNA Are Localized at the Plasma Membrane and the Regions of Moving Cytoplasm during the Cellular Response to Injury. The Journal of Cell Biology, Volume 112.

  1. The nature of the adhesive film should be provided. If commercially available, please provide the information. What is the difference between the Kawamoto’s Film Method and this?

The nature of the adhesive film has been added (see line 102).

There are several differences between the cryosection method developed by Kawamoto et al. (1990, 2003, 2021) and the one developed in our laboratory. 1. We use a different type of film (39475214, Leica). 2. Section treatments were performed in the film and then transferred to slide. For the transfer they used a special solution. In our method, the sections were transferred immediately to the slide and then the different treatments (staining, immunofluorescence and ISH) were performed directly on it. For the transfer we used UV light. This step is very important because as a consequence, the cryosections can be stored at -80 degree for a later use without RNA degradation. In addition, at antigen retrieval for immunolabellingthe samples are treated at 121 celsius. Using our method this treatment can be safly performed. 3. In the Kawamoto et al. (1990, 2003, 2021) work the authors did not analyzed the cryosection with in situ-HCR technique which is an isothermal enzyme-independent nucleotide polymerization method. The concept of this method is to use two hairpin oligonucleotides linked with fluorophores. When the complementary initiator nucleotide hybridizes with mRNA, the initiator activates the hairpin molecules, and they assemble into a well-defined structure providing a source of fluorescence. The main advantage of this method is that it is very sensitive and therefore suitable for the detection of the low abundant mRNAs.

  1. At least the authors should quote the Kawamoto reference.

Kawamoto T, Kawamoto K. Preparation of Thin Frozen Sections from Nonfixed and Undecalcified Hard Tissues Using Kawamoto's Film Method (2020). Methods Mol Biol. 2021;2230:259-281. doi: 10.1007/978-1-0716-1028-2_15. PMID: 33197019.

Thank you for your remark, the reference has been added (see lines 68, 450-455).

Reviewer 2 Report

The improvement of methods to study bone tissue during development, injury and regeneration is an important aspect for musculoskeletal research. Cryosectioning of hard tissue has been introduced several decades ago (e.g. PMID: 387879) and optimized by the tape technique by Kawamoto (PMID: 2254645 PMID: 12846553, PMID: 33197019). The authors newer mentioned Kawamoto in the manuscript. The CryoJane tape transfer system is now commercially available. The new aspect of the presented method sems to be that no fixation and decalcification of the tissue was done before sectioning. This, however, is not well highlighted in the manuscript. The use of actin for ISH needs further explanation, as this is a very general protein of the cytoskeleton. The manuscript would benefit for the analysis of more “bone specific” probes. Furthermore, a comparison with the standard methods might show if the presented method is beneficial compared to the previous methods.

Introduction:

  1. The meaning of these sentences is not correct or unclear to me: “Gene expression analysis of the bone cells mostly uses immunofluorescence technique methods for the detection of proteins in situ [11]. In spite of immunolabeling being very advantageous and well reproducible, there are some disadvantages, such as non-specific labeling with certain antibodies [12,13]. Additional difficulty in expression analysis in situ is the necessity to identify the cytokines they secrete.” 1. Gene expression cannot be analyzed by IHC /detection of proteins. 2. To analyse gene expression, always specific primer or sequences of specific genes are used.
  2. They should also mention FISH in the introduction.
  3. The statement that paraffin embedded bone is not appropriate for ISH is not correct and should be corrected. E.g. Ref#26 states “In our opinion, fixation in buffered formalin, decalcification in EDTA, and embedding in paraffin is a good compromise, because it not only provides good morphology and excellent conditions for immunohistochemistry but also allows DNA- and RNA- based molecular studies.”
  4. The use of cryosections for ISH and a detailed protocol has already been described in Ref 29, 30. What is the new information given by this technical note?

Materials & Methods

  1. What is the Buffer A? If they use the standard protocol of the CryoJane Tape-transfer system, this should be mentioned. Please cite relevant studies, where this system has already been described.
  2. 1M. Except for the first picture, the differences of the following 3 is not so obvious between the upper and lower row. This is confusing. Please explain the technical practices that should be avoided.
  3. All abbreviations must be spelled out at first use, e.g. HES.
  4. HES stain: looking at Fig 1j-k I can hardly see a yellow stain. However, based on the description of the staining, collagen should be stained in yellow.
  5. ISH: Why was actin chosen? The authors should show the labeling with more tissue specific probes, such as RUNX, ALP, Col10 for chondrocytes or osteoclast markers.
  6. 6.4 Which fluorophore?

Results:

  1. Please explain the distribution of the actin labelling. Actin, as part of the cytoskeleton, should be detectable in all osteoblasts, osteocytes, osteoclast, chondrocytes, fibroblast, etc. Please show more specific pictures.
  2. Please mention the counterstain in the in situ HCR.
  3. A comparison of ISH on decalcified and non-decalcified, cryo-sections and paraffin-sections would be of great interest.
  4. A table comparing the previous methods and the newly proposed would be beneficial.

Conclusion:

  1. What do they mean with this sentence: “For proper regeneration, permanent relationship between external and internal signals is indispensable.”?
  2. IRB: Minipigs? They used rat tissue.

Author Response

Dear Reviewer 2,

The Reviewer 2 comments greatly helped to improve our manuscript. We have modified the manuscript by taking into account each comment and remark. To improve the manuscript additional experimental data have been included (Fig. 2, Fig. 3, Fig. 5, Fig. 6, Table 1 and Table S1).

The improvement of methods to study bone tissue during development, injury and regeneration is an important aspect for musculoskeletal research. Cryosectioning of hard tissue has been introduced several decades ago (e.g. PMID: 387879) and optimized by the tape technique by Kawamoto (PMID: 2254645 PMID: 12846553, PMID: 33197019). The authors never mentioned Kawamoto in the manuscript.

 The manuscript has been modified and the reference added (see lines 68, 450-455).

The CryoJane tape transfer system is now commercially available. The new aspect of the presented method sems to be that no fixation and decalcification of the tissue was done before sectioning. This, however, is not well highlighted in the manuscript.

This point has been highlighted in the manuscript (see lines 68-69).

The use of actin for ISH needs further explanation, as this is a very general protein of the cytoskeleton.

Thank you for your remark. Beta-actin is often used as positive control in in situ hybridization*, because it is highly expressed in almost every cell.

*F Y Gan, G D Luk and M S Gesell. (1994) Nonradioactive in situ Hybridization Techniques for Routinely Prepared Pathology Specimens and Cultured Cells. Journal of Histotechnology Volume 17 313-319 | Published online: 29 Nov 2013.

K L Taneja and R H Singer. (1990) Detection and localization of actin mRNA isoforms in chicken muscle cells by in situ hybridization using biotinated oligonucleotide probes. J Cell Biochem, 1990 Dec;44(4):241-52. doi: 10.1002/jcb.240440406.

T C Hoock, P M Newcomb, and I M Herman. (1991) Beta-Actin and its mRNA Are Localized at the Plasma Membrane and the Regions of Moving Cytoplasm during the Cellular Response to Injury. The Journal of Cell Biology, Volume 112.

The manuscript would benefit for the analysis of more “bone specific” probes.

The manuscript has been modified to improve the message of the paper (see lines 37-43, 135-143, 240-243, 287-295, 312-319, 329-330). Additional experimental data have been included (Fig. 2, Fig. 3, Fig. 5, Fig. 6 and Table S1).

Furthermore, a comparison with the standard methods might show if the presented method is beneficial compared to the previous methods.

 The manuscript has been modified (see lines 325-327 and Table 1).

Introduction:

  1. The meaning of these sentences is not correct or unclear to me: “Gene expression analysis of the bone cells mostly uses immunofluorescence technique methods for the detection of proteins in situ [11]. In spite of immunolabeling being very advantageous and well reproducible, there are some disadvantages, such as non-specific labeling with certain antibodies [12,13]. Additional difficulty in expression analysis in situ is the necessity to identify the cytokines they secrete.” 1. Gene expression cannot be analyzed by IHC /detection of proteins. 2. To analyse gene expression, always specific primer or sequences of specific genes are used.

The text has been changed (see lines 45-46, 48-49).

  1. They should also mention FISH in the introduction.

The FISH mention has been added (see lines 57-60).

  1. The statement that paraffin embedded bone is not appropriate for ISH is not correct and should be corrected. E.g. Ref#26 states “In our opinion, fixation in buffered formalin, decalcification in EDTA, and embedding in paraffin is a good compromise, because it not only provides good morphology and excellent conditions for immunohistochemistry but also allows DNA- and RNA- based molecular studies.”

Thank you for your remark, the text has been corrected (see lines 64-66).

  1. The use of cryosections for ISH and a detailed protocol has already been described in Ref 29, 30. What is the new information given by this technical note?

In the Kramer et al. (2012) protocol the ISH has been performed in the paraffin embedded bone section. In the Salie et al. (2008) papert the authors analyzed the cryosection only with in situ-DIG technique and not with in situ-HCR technique. HCR technique which is an isothermal enzyme-independent nucleotide polymerization method. The concept of this method is to use two hairpin oligonucleotides linked with fluorophores When the complementary initiator nucleotide hybridizes with mRNA, the initiator activates the hairpin molecules, and they assemble into a well-defined structure providing a source of fluorescence. The main advantage of this method is that it is very sensitive and therefore suitable for the detection of the low abundant mRNAs.

There are several differences between the cryosection method developed by Kawamoto et al. (1990, 2003, 2021) and the one developed in our laboratory. 1. We use a different type of film (39475214, Leica). 2. Section treatments were performed in the film and then transferred to slide. For the transfer they used a special solution. In our method, the sections were transferred immediately to the slide and then the different treatments (staining, immunofluorescence and ISH) were performed directly on it. For the transfer we used UV light. This step is very important because as a consequence, the cryosections can be stored at -80 degree for a later use without RNA degradation. In addition, at antigen retrieval for immunolabellingthe samples are treated at 121 celsius. Using our method this treatment can be safly performed.

What is the Buffer A? If they use the standard protocol of the CryoJane Tape-transfer system, this should be mentioned. Please cite relevant studies, where this system has already been described.

Standard solution of the the CryoJane Tape-transfer system was used. 5 ml of A buffer concentrate (39475270, Leica), 1.25 ml of 0.1 M acetic acid and 25 ml of acetone in 500 ml A solution. The text has been corrected (see lines 86-87).

  1. 1M. Except for the first picture, the differences of the following 3 is not so obvious between the upper and lower row. This is confusing. Please explain the technical practices that should be avoided.

Text has been added to improve the understanding of the images (see lines 110-114).

  1. All abbreviations must be spelled out at first use, e.g. HES.

The text has been corrected (see line 116).

  1. HES stain: looking at Fig 1j-k I can hardly see a yellow stain. However, based on the description of the staining, collagen should be stained in yellow.

We apologize for this error. We used Hematoxylin and phloxin (HP) stain. The error has been corrected (see lines 116, 284).

  1. ISH: Why was actin chosen? The authors should show the labeling with more tissue specific probes, such as RUNX, ALP, Col10 for chondrocytes or osteoclast markers.

The expression of beta-actin is uniform in almost every cells. Therefore, beta-actin is a useful positive control of in situ hybridization and it is frequently used as a positive control in publications.* Its strong and “ubiquitous” expression is an advantage.

*F Y Gan, G D Luk and M S Gesell. (1994) Nonradioactive in situ Hybridization Techniques for Routinely Prepared Pathology Specimens and Cultured Cells. Journal of Histotechnology Volume 17 313-319 | Published online: 29 Nov 2013.

K L Taneja and R H Singer. (1990) Detection and localization of actin mRNA isoforms in chicken muscle cells by in situ hybridization using biotinated oligonucleotide probes. J Cell Biochem, 1990 Dec;44(4):241-52. doi: 10.1002/jcb.240440406.

T C Hoock, P M Newcomb, and I M Herman. (1991) Beta-Actin and its mRNA Are Localized at the Plasma Membrane and the Regions of Moving Cytoplasm during the Cellular Response to Injury. The Journal of Cell Biology, Volume 112.

The remark has been taken into account and additional experimental data have been included (Fig. 2, Fig. 3, Fig. 5, Fig. 6, Table 1 and Table S1). The manuscript has been modified to improve the message of the paper (see lines 37-43, 135-143, 240-243, 287-295, 312-319, 329-330).

  1. 6.4 Which fluorophore?

The manuscript has been modified (see line 249). Alexa Fluor488

Results:

  1. Please explain the distribution of the actin labelling. Actin, as part of the cytoskeleton, should be detectable in all osteoblasts, osteocytes, osteoclast, chondrocytes, fibroblast, etc. Please show more specific pictures.

As commented on by the reviewer, the expression of beta-actin is uniform in almost every cell. Therefore, beta-actin is a useful positive control of in situ hybridization and it is frequently used as a positive control in publications.*

*F Y Gan, G D Luk and M S Gesell. (1994) Nonradioactive in situ Hybridization Techniques for Routinely Prepared Pathology Specimens and Cultured Cells. Journal of Histotechnology Volume 17 313-319 | Published online: 29 Nov 2013.

K L Taneja and R H Singer. (1990) Detection and localization of actin mRNA isoforms in chicken muscle cells by in situ hybridization using biotinated oligonucleotide probes. J Cell Biochem, 1990 Dec;44(4):241-52. doi: 10.1002/jcb.240440406.

T C Hoock, P M Newcomb, and I M Herman. (1991) Beta-Actin and its mRNA Are Localized at the Plasma Membrane and the Regions of Moving Cytoplasm during the Cellular Response to Injury. The Journal of Cell Biology, Volume 112.

More specific labeling with anti-Cd68 antibody has been added (see Figure 3 and 5).

  1. Please mention the counterstain in the in situ HCR.

DAPI has been used as a nuclear counterstain. The mention of the counterstain has been added (see line 309).

  1. A comparison of ISH on decalcified and non-decalcified, cryo-sections and paraffin-sections would be of great interest.

The Table 1 has been added and the text has been modified (see lines 64-69, 325-327).

  1. A table comparing the previous methods and the newly proposed would be beneficial.

The Table 1 has been added.

Conclusion:

  1. What do they mean with this sentence: “For proper regeneration, permanent relationship between external and internal signals is indispensable.”?

To clarify this point, the text has been modified to improve the understanding (see lines 336-337).

  1. IRB: Minipigs? They used rat tissue.

We apologize for this error. The error has been corrected. (see line 366).

Round 2

Reviewer 1 Report

The revised version includes osteoclast/macrophage data as well as the detailed explanation on the film method. 

Reviewer 2 Report

The authors have extensively revised the manuscript. They have responded to all the comments raised, and added tables, data, stainings and other information. This improved the manuscript.